# Long-Term Survival Analysis and Prognostic Factors of Arabic Patients with Differentiated Thyroid Carcinoma: A 20-Year Observational Study at the King Hussein Cancer Center (KHCC) Involving 528 Patients

**DOI:** 10.3390/cancers15164102

**Published:** 2023-08-15

**Authors:** Akram Al-Ibraheem, Ula Al-Rasheed, Noor Mashhadani, Ahmed Saad Abdlkadir, Dhuha Ali Al-Adhami, Saad Ruzzeh, Feras Istatieh, Areen Mansour, Basem Hamdan, Reem Kheetan, Marwa Al-Shatti, Issa Mohamad, Malik E. Juweid, Areej Abu Sheikha, Kamal Al-Rabi, Gerasimos P. Sykiotis, Michael C. Kreissl, Taleb Ismael, Iyad Sultan, Hikmat Abdel-Razeq

**Affiliations:** 1Department of Nuclear Medicine and PET/CT, King Hussein Cancer Center (KHCC), Al-Jubeiha, Amman 11941, Jordan; 2Department of Radiology and Nuclear Medicine, Division of Nuclear Medicine, University of Jordan, Al-Jubeiha, Amman 11942, Jordan; 3Office of Scientific Affairs and Research, King Hussein Cancer Center (KHCC), Al-Jubeiha, Amman 11941, Jordan; 4Department of Surgery, King Hussein Cancer Center (KHCC), Al-Jubeiha, Amman 11941, Jordan; 5Department of Medicine, King Hussein Cancer Center (KHCC), Al-Jubeiha, Amman 11941, Jordan; 6Department of Pathology and Laboratory Medicine, King Hussein Cancer Center (KHCC), Al-Jubeiha, Amman 11941, Jordan; 7Department of Radiation Oncology, King Hussein Cancer Center (KHCC), Al-Jubeiha, Amman 11941, Jordan; 8Department of Endocrinology, Diabetology and Metabolism, Vaud University Hospital Center (CHUV), CH-1011 Lausanne, Switzerland; 9Department of Radiology and Nuclear Medicine, University Hospital of Magdeburg, 39120 Magdeburg, Germany; 10Department of Pediatrics, King Hussein Cancer Center (KHCC), Al-Jubeiha, Amman 11941, Jordan

**Keywords:** thyroid cancer, differentiated thyroid cancer, radioactive iodine, papillary thyroid cancer, KHCC, prognostic factors

## Abstract

**Simple Summary:**

The impact of ethnic and geographic variation on observed differentiated thyroid cancer (DTC) outcomes has been well-documented, demonstrating distinct and disparate observations across different countries. However, there remains a paucity of evidence regarding the analysis of factors influencing DTC outcomes, specifically within the Arabic population. This study examines the long-term outcome of DTC in the Arab population, looking at various influential factors. The analysis of 528 cases revealed favorable survival outcomes. Disease progression was observed in a subset of cases, but the majority of patients achieved positive outcomes. Age at diagnosis, gender, risk categorization, and tumor stage impact both disease progression and mortality. The survival rates were favorable for patients who received RAI treatment. Moreover, adherence to institutional clinical practice guidelines (CPG) resulted in better disease control. Our study features the largest Middle Eastern and Arab cohort to date, highlighting the long-term favorable outcomes of DTC patients.

**Abstract:**

DTC accounts for the majority of endocrine tumors. While the incidence of thyroid cancer has been increasing globally over the past few decades, papillary thyroid carcinoma (PTC) generally shows an excellent prognosis, except in cases with aggressive clinicopathological features. This study aimed to assess the 5- and 10-year overall survival (OS) and progression-free survival (PFS) of 528 Arabic patients diagnosed with primary DTC from 1998 to 2021. Additionally, the study aimed to analyze the impact of various factors on both OS and PFS. An univariable survival analysis was conducted using Kaplan–Meier curves. The 5- and 10-year OS for patients with DTC have exceeded 95%. Additionally, PFS showed very good rates (ranging between 96.5 and 85% at 5 and 10 years, respectively). Age, male gender, risk of recurrence, and distant metastasis were identified as the main negative prognostic factors for both OS and PFS, while RAI treatment was found to be a significant factor in improving OS. Moreover, adherence to the King Hussein Cancer Center’s (KHCC) CPG demonstrated significant improvement in PFS. These findings highlight common prognostic factors and favorable outcomes in Arabic patients with DTC treated at a tertiary cancer center using standard of care approaches.

## 1. Introduction

DTC is the most common neoplasm of the endocrine system; it accounts for 90% of endocrine tumors, 3.1% of all cancers, and 1% of all newly diagnosed malignancies [1,2].

Histopathologically, thyroid malignancies are divided into PTC (80%), follicular carcinomas (10%), medullary thyroid carcinomas (5–10%), poorly differentiated thyroid carcinoma (2–15%), and anaplastic carcinomas (1–2%) [3]. Hürthle cell carcinoma is a rare thyroid malignancy that is often considered a variant of follicular carcinoma. According to the 2022 classification from the World Health Organization (WHO), Hürthle cell carcinomas account for 2–3% of all thyroid malignancies [3].

Thyroid cancer management is composed of both surgical (total or hemi-thyroidectomies associated with central and lateral neck dissection) and adjuvant medical treatment such as RAI, thyroxine and tyrosine kinase inhibitors (TKI), or chemotherapy [4,5,6,7]. In general, the prognosis of well-differentiated thyroid cancers is believed to be excellent, particularly for localized DTC and thyroid cancer with regional lymph node metastases [8]. To our knowledge, there is only limited research work on prognostic factors and outcomes of DTC, specifically for Middle Eastern and Arab populations [9,10,11,12].

The global incidence of different types of DTC has significantly increased over the past four decades; the average increase was 48.0% among males and 66.7% among females [13]. More recently, the age-adjusted international thyroid cancer incidence rates from 1998 to 2002 varied 5-fold for males and nearly 10-fold for females by geographic region [13].

Overall, the long-term impact of effective treatment of DTC is excellent in most patients [14]. One major factor is that the majority of the increase in incidence is in histological types with a favorable prognosis, primarily DTC and early-stage disease [15,16]. Nonetheless, there is still a small proportion of patients showing recurrence, and some patients do not respond to conventional treatment and can eventually die from DTC [11].

Predicting long-term prognosis (including potential death) based on treatment options, histopathological diagnosis, tumor classification, and different sociodemographic factors is still challenging for this common type of neoplasm. Nonetheless, because DTC is associated with a low mortality rate and requires long-term and demanding follow-up, conventional prospective randomized controlled trials related to treatment options, like RAI to identify factors that predict the long-term prognosis of DTC, are generally very difficult.

Herein, we present a large retrospective cohort of DTC patients who were diagnosed, treated, and sufficiently followed-up at the KHCC and identify various influential factors on long-term outcomes.

## 2. Materials and Methods

### 2.1. Patient Selection

The present study analyzed data collected retrospectively from a tertiary cancer center. The cohort under investigation comprised 563 patients who received a diagnosis of thyroid cancer through cytologic or histopathologic confirmation. The data collected during the period between September 1998 and December 2021 were retrospectively analyzed. A total of 528 patients were identified as having DTC (Table 1). Notably, patients with other undifferentiated histopathologic subtypes were omitted from the analysis. Additionally, individuals with incomplete medical records that lacked important pre-determined parameters (n = 23) were also excluded from the study.

The Ethics Committee of the KHCC Institute examined and granted approval for this research project (approval reference 22 KHCC 131, 11 August 2022). A retrospective examination was performed on the medical records of patients to evaluate various characteristics. These included factors such as age at the time of diagnosis, gender, and histologic subtypes of the tumor. Primary tumor size and staging categories for the T, N, and M staging categories were also examined. Assessment of TNM staging was examined using the classification guidelines of the American Joint Committee on Cancer (AJCC) 8th edition [17]. Additionally, the risk of recurrence was stratified according to the 2015 guidelines of the American Thyroid Association (ATA) [18]. Patterns of response after each therapeutic or diagnostic scan were determined by reviewing RAI whole-body images. Furthermore, neck ultrasound (US) and the laboratory findings of serum thyroglobulin (sTg), and thyroglobulin antibodies (TgAb) were also reviewed for response assessment.

### 2.2. Procedures for Therapeutic Radioiodine Administration

The average interval between total thyroidectomy and initial RAI was 75.6 days (ranging between 23 and 217). The referral of new patients to our department was determined by the decision of the multidisciplinary clinic (MDC) following surgery. Patients were asked to withdraw from levothyroxine therapy four weeks before the planned administration of therapeutic RAI.

Recombinant human thyroid-stimulating hormone (TSH) was only used in patients with comorbidities who could not tolerate withdrawal and in cases of non-increasing TSH despite withdrawal. Patients were also instructed to avoid iodine-rich foods, not to undergo contrast-enhanced computed tomography (CT), and to follow a low-iodine diet (200 µg per day) for 2–3 weeks before RAI and four days after the administration of RAI. Hospital admission was mandatory for patients receiving more than 1121 MBq of RAI for radiation protection purposes. These patients were kept in isolation in the medical ward for 24 h until the residual radioactivity in their bodies decreases to 500 MBq or less [19]. Whole-body scintigraphy (WBS) was then performed to examine the biodistribution of RAI.

### 2.3. Procedure for Radioiodine WBS

Four to ten days after oral administration of therapeutic RAI, a standard WBS was performed using a whole-body moving-camera technique (anterior and posterior). Dual-head gamma cameras with high-energy collimators were used. WBS images were acquired in planar view with a pair of appropriate window levels (dark and bright). When a positive finding was suspected, an oblique view or single photon emission computed tomography (SPECT) images (axial and coronal) were obtained additionally. The RAI uptake in post-therapeutic WBS was evaluated using a 3-grade system in which 0 is an excellent response, 1 is a partial response, and 2 is poor [19]. The diagnostic RAI whole-body scan was performed 4–6 months later using oral RAI dosed between 74 and 185 MBq. The results of the diagnostic WBS were correlated with neck US and sTg results, which were obtained after adequate thyroxine deprivation. A grade of 0 on subsequent diagnostic whole-body scan (WBS), neck ultrasound (US), and markers’ profile indicates no observed uptake of RAI in the thyroid gland. Patients in the grade 1 category still show evidence of uptake in the disease, which is partially improved compared to their pre-therapeutic profile in terms of diagnostic WBS, neck US, and markers [19]. However, a grade of 2 signifies a substantial uptake of RAI in the thyroid gland, either showing progression or stabilization since previous profiles, indicating non-response.

### 2.4. Late Follow-Up

The follow-up process included a thorough clinical examination, testing thyroid function, and measuring sTg and TgAb. Patients also underwent neck US scans, chest radiographs, and diagnostic WBS every 6 to 12 months during the first 2 years after surgery. Results retrieved from all aforementioned entities were examined and classified. In an attempt to improve interdisciplinary communication and establish consistent management protocols for patients with DTC, a CPG was created and extensively utilized at the KHCC over the past decade. These guidelines were the result of productive collaboration among various departments, including nuclear medicine, endocrinology, surgical oncology, medical oncology, radiation oncology, and pathology [20]. The CPG undergoes annual revisions to ensure that physicians are equipped with current information and best practices in all aspects of DTC. The individuals who were exposed to this contemporary approach, involving 184 patients (46 males and 138 females), underwent a comprehensive evaluation and analysis, in contrast to a separate cohort of patients (n = 344) who were treated using traditional methods. The aim was to evaluate the potential merits and advantages of this approach.

### 2.5. Statistical Analysis

Baseline statistics were reported using measures of central tendency and dispersion for continuous variables and proportions for categorical variables. The Kaplan–Meier method was used to determine OS and PFS. In the case of patients who were lost to follow-up or those who died from a cause unrelated to thyroid cancer, the date of the last evaluation was used. A log-rank test was used to compare the survival curves. All tests were two-sided and a *p* value of less than 0.05 was considered statistically significant. SPSS 29.0 for Windows (SPSS, Inc., Chicago, IL, USA) was used to perform the above-mentioned analyses. Only statistically significant factors are thereafter reported and discussed.

## 3. Results

### 3.1. Patient Characteristics

A total of 528 patients with DTC were retrospectively enrolled. The median follow-up duration in all patients with DTC was 153 months (range, 13–198 months). The majority of patients were female (72%), while only 28% were male. The average age at diagnosis was 39 (range, 4–88). About 84.5% of patients were younger than 55 years old and only 2.3% were below the age of 16. The mean tumor size was 2.7 cm (range, 0.1–15 cm). In total, 39.3% of patients had multifocal tumors, and 26.2% of patients had identifiable extrathyroidal extensions. Cervical lymph node metastases were observed histopathologically in a total of 226 patients (21% with N1a and 29.4% with N1b).

### 3.2. Treatment Modalities

Among all 528 patients, 438 (83%) underwent total thyroidectomy upon initial detection of DTC. Partial thyroidectomy (either lobectomy or lumpectomy) was approached first for 32 patients following their initial presentation with suspicious thyroid nodules. Subsequently, complete thyroidectomy was adopted for those patients upon the identification of DTC. A minority of patients (n = 39) received either palliative therapy or TKI due to advanced presentation and subsequent dedifferentiation (Table 1).

### 3.3. Survival Outcome

Among all 528 cases, only 24 patients died from DTC during the median follow-up period of 153 months. An additional 14 other patients who died from non-disease-specific reasons (ischemic heart disease, sepsis, respiratory failure, breast cancer, and renal failure due to diabetes) were censored in the present study. Disease progression was evident in 135 cases. The 5-year OS and PFS were found to be 97.9% and 95%, respectively (Figure 1).

The mean age at diagnosis for deceased patients was 58 years (range, 28–77 years), which was significantly greater than that of survivors (*p* value < 0.01). There were no deceased patients younger than 28 years old at the time of initial RAI, and no deceased patients were observed with stage I disease. Based on the cohort data, the 10-year OS was 96.5%, whereas the 10-year PFS was 85% (Figure 1). The mean estimated survival time (ST) for all patients within the cohort was 140.1 months, while the mean progression-free time (PFT) was 129.4 months.

Our survival analysis revealed a statistically significant association between age ≥ 55 years and a worse PFT and ST (116.7 months and 139.3 months, respectively, *p* < 0.01). Conversely, younger patients exhibited a significantly more favorable PFT and ST, with approximately 173.7 months and 196 months, respectively (Figure 2).

The mean estimated ST in females was 194.4 months (Figure 2A), which was significantly higher than the mean estimated ST in males, (156.4 months, *p* = 0.02). Similarly, the mean estimated PFT has also demonstrated a statistical difference (Figure 2B) in favor of the female gender (172.9 months vs. 134.4 months, *p* < 0.01).

Furthermore, patients who received RAI treatments showed a significantly higher mean ST compared to patients who did not receive RAI treatments (*p* = 0.01).

Patients with a high-risk category at the time of diagnosis showed a lower estimated mean ST of 149 months when compared to patients with low and intermediate risks. Low- and intermediate-risk patients had a more favorable survival outcome, as demonstrated by higher and statistically significant estimated mean ST of about 166 months (*p* < 0.01, Figure 2A). The difference in the ATA risk groups was more clearly demarcated when calculating PFT. Low-risk patients were the best group that achieved a mean ST of 157.4 months, followed by the intermediate-risk group. Intermediate-risk group patients also had a much better ST than the high-risk groups (142.3 vs. 110.2 months). The difference between each risk group was statistically significant (*p* < 0.01, Figure 2B).

Finally, patients with distant metastasis at the time of diagnosis showed a lower mean estimated ST and PFT (143 and 106 months, *p* < 0.01 each). It is noteworthy that a subgroup of patients who were treated according to our CPG involving 184 patients (46 males and 138 females) (179 alive and 5 dead) showed a better PFS (*p* value < 0.01), demonstrating 100% and 91.9% 5- and 10-year rates, respectively (Table 2, Figure 2B). Table 2 demonstrates the significance of RAI and other patient-related factors on PFT and ST (Table 2).

## 4. Discussion

Ethnic variation in thyroid cancer has received significant acknowledgment in the existing literature. Many ethnic groups, races, and nations display deviations from the established trends and factors influencing survival rates [21]. For Arab and Middle Eastern populations, this study will serve as a reference observation looking at various factors affecting DTC outcomes on long-term follow-up. A female predominance was noted, endorsing similar observations from Egypt, United Arab Emirates, Lebanon, and Yemen [22,23,24,25]. The average age of onset for thyroid cancer among Arabs was observed to occur during the third decade of life. This is approximately ten years earlier compared to findings from other global studies [8,18]. PTC is the predominant histologic subtype similar to worldwide observations acknowledged in iodine-sufficient regions [26]. For Arab patients, previous studies that examined the significance of gender on DTC survival were divergent [22,23,24,25]. In fact, some studies found that gender disparity has no effect on DTC survival at all [23]. In our study, the female gender was linked to more favorable survival times.

Factors that are showing the most significant difference in the survival of patients with DTC are the RAI therapy, the higher age at diagnosis, and the presence of distant metastasis, as shown for patients with stage IV tumors. Thus, the main negative influencers for prognosis in patients with DTC are age ≥ 55 years (AJCC 8th edition) [27] and the presence of distant metastasis. Younger patients, patients with localized tumors, and patients who receive RAI therapy have a significantly better prognosis. Compared to their male counterparts, female patients had a significantly better survival rate with a 96% survival rate compared to 87.3% for men. This finding is supported by a prospective study conducted on patients with DTC who were treated at the U.S. Air Force or Ohio State University hospitals [28]. The study concluded that being female significantly decreased the likelihood of cancer-related mortality [28].

The low mortality rate of the DTC in general, which is also reported herein, makes the evaluation of prognostic factors related to patient and tumor and the efficiency of treatment options more difficult to assess compared to other survival analysis studies. In a study by Borges et al., a cohort of patients with DTC in Brazil showed a 5-year disease-related OS of about 95.8% [15]. In another retrospective review of 269 patients with PTC, it was observed that older age, metastases, and stages ≥ III were predictors of death from PTC [29]. Our study suggests clearly that the two most significant factors that are associated with lower expected survival in patients with DTC are the presence of distant metastasis (or stage IV cancer) and the higher age at the time of diagnosis (age ≥ 55 years as per TNM AJCC 8th edition).

The value of RAI ablation is a controversial topic, mainly for patients who underwent radical thyroidectomy, low-risk patients, and patients among the younger age groups due to the associated risk of a second primary neoplasm [30,31,32]. There is a clear consensus among NM physicians regarding the routine use of RAI for high-risk and low-risk patient groups. However, a significant challenge arises when considering a large group of patients who fall into the intermediate-risk category. This intermediate-risk group has proven to be challenging in both our study and previous research for several reasons [33]. Firstly, these patients generally have positive outcomes, with low mortality rates and a limited number of recurrences. Additionally, the majority of these patients are eligible for RAI and ultimately receive it. Thus, it is difficult to retrospectively analyze the impact of RAI in this group since nearly all patients undergo RAI treatment. Moving forward, it is hoped that ongoing research will shed more light on the impact of RAI in this intermediate-risk group. Prospective evidence and long-term studies on the effects of RAI in intermediate-risk patients are eagerly awaited as they may offer valuable insights into the role of radioiodine ablation in this patient population [33].

Beyond guideline-based risk stratifications, clinicians and healthcare providers in Asia consider various clinico-social factors when selecting the RAI dose for low-risk DTC patients, according to a survey study [34]. The results indicate that, in addition to following guidelines, personalized care is essential in the management of DTC, and clinicians should also take into account patients’ clinical and socioeconomic conditions [34].

This study has some limitations, including its retrospective nature and single-center experience. Nonetheless, it remains one of a few studies that examine the impact of various influential factors among Arabian and Middle Eastern DTC patients.

## 5. Conclusions

Our cohort of DTC Arab patients at the KHCC had a high OS and PFS, which endorses evidence from the literature in different ethnic populations. The prognosis was negatively influenced by the male gender, ≥55-year age, ATA high-risk category, and TNM metastatic status, whereas RAI therapy was found to be effective in prolonging ST. Moreover, DTC patients who were treated according to the KHCC CPG have achieved an optimal PFS (100% over 5 years and 91.9% over 10 years) when compared to the whole cohort. These findings suggest that appropriate treatment and adherence to guidelines can improve outcomes for DTC patients.

## Figures and Tables

**Figure 1 cancers-15-04102-f001:**
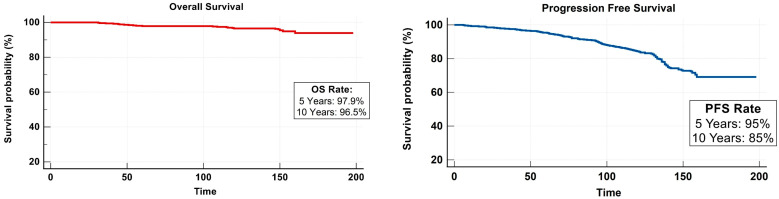
Kaplan–Meir plots denoting survival analysis for both OS and PFS of the study cohort.

**Figure 2 cancers-15-04102-f002:**
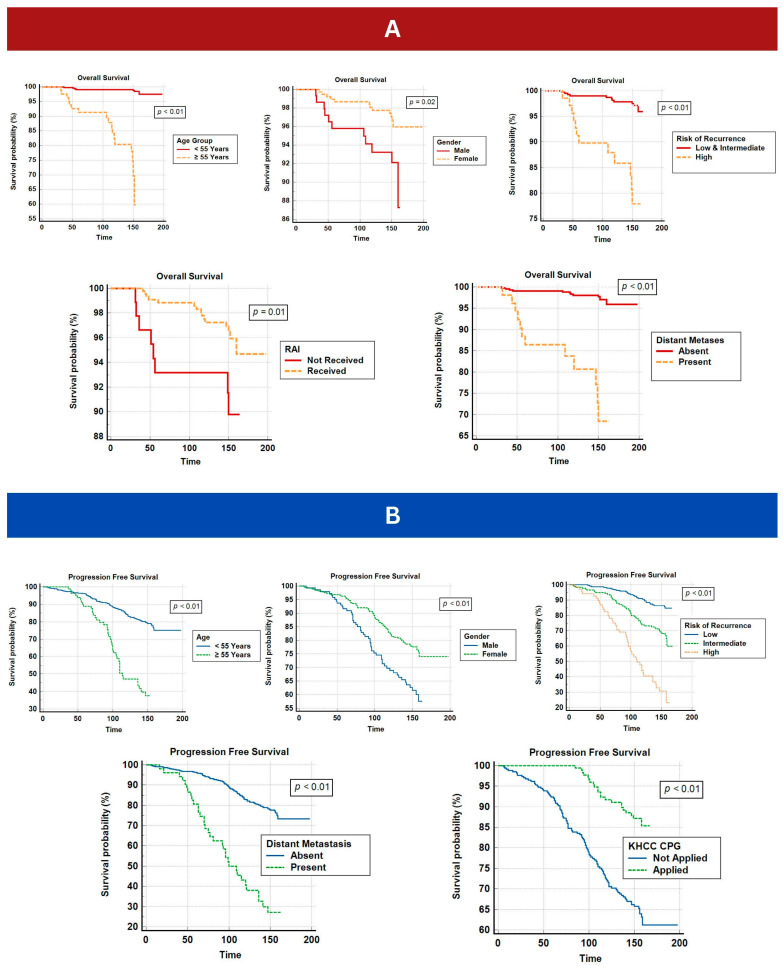
(**A**) Kaplan–Meir plots denoting survival analysis for OS in relation to variable significant clinicopathologic characteristics. (**B**) Kaplan–Meir plots denoting survival analysis for PFS in relation to variable significant clinicopathologic characteristics.

**Table 1 cancers-15-04102-t001:** Demographic, histopathological, and clinical characteristics of study sample.

**Demographics**	
**Age At Diagnosis (in Years)**	
Mean	39 years
Range	4–88 years
**Gender (Number, Percentage)**	
Male	148 (28%)
Female	380 (72%)
**Nationality**	
Jordan	354 (67%)
Palestine	40 (7.6%)
Iraq	33 (6.2%)
Libya	31 (5.9%)
Egypt	26 (4.9%)
Yemen	22 (4.2%)
Syria	17 (3.2%)
Sudan	5 (1%)
**Histopathological Characteristics**	
**Histologic Subtypes (Number, Percentage)**	
Papillary	468 (88.6%)
Follicular	40 (7.6%)
Hürthle	20 (3.8%)
**Tumor Size (cm mean and range)**	2.7 cm (0.1–15 cm)
**T-Category (Number, Percentage)**	
T1a	57 (10.8%)
T1b	120 (22.7%)
T2	149 (28.2%)
T3a	91 (17.3%)
T3b	66 (12.5%)
T4a	24 (4.5%)
T4b	21 (4%)
**N-Category (Number, Percentage)**	
N0	262 (49.6%)
N1a	111 (21%)
N1b	155 (29.4%)
**M-Category (Number, Percentage)**	
M0	470 (89%)
M1	58 (11%)
**Overall Staging (Number, Percentage), according to AJCC ^1^ 8th edition**	
Stage I	349 (66.1%)
Stage II	73 (13.8%)
Stage III	46 (8.7%)
Stage IVa	38 (7.2%)
Stage IVb	14 (2.7%)
Stage IVc	8 (1.5%)
**Treatment Modality (Number, Percentage)**	
Total Thyroidectomy	438 (83%)
Partial Thyroidectomy	32 (6%)
Completion thyroidectomy	32 (6%)
RAI ^2^	431 (81.6%)
Tyrosine Kinase inhibitors	10 (2.6%)
Palliative Therapy	29 (5.5%)
**RAI Treatment Received (Number, Percentage)**	
Single Dose	344 (65.2%)
Multiple Doses	94 (17.8%)
Not Received	90 (17%)
**Response after RAI Therapy (Number, Percentage)**	
Complete Response	316 (59.8%)
Partial Response	117 (21.2%)
No Response	5 (1%)
**Cumulative RAI dose (GBq mean and range)**	5.5 GBq (1.1–37 GBq)
**Risk of Recurrence (Number, Percentage), according to 2015 ATA ^3^ guidelines**	
Low Risk	257 (48.7%)
Intermediate Risk	190 (36%)
High Risk	81 (15.3%)

^1^ AJCC: American Joint Committee on Cancer; ^2^ RAI: Radioactive Iodine; ^3^ ATA: American Thyroid Association.

**Table 2 cancers-15-04102-t002:** A table summary demonstrating the significance of RAI and other patient-related factors on PFT and ST.

Variable		PFT ^1^	ST ^2^	*p* Value
Age at diagnosis	<55 Years	173.7	196	<0.01 (For PFS)
	≥55 Years	116.7	139.3	<0.01 (For OS)
Gender	Female	172.9	194.4	<0.01 (For PFS)
	Male	134.4	156.4	0.02 (For OS)
Risk of recurrence	Low	157.4		<0.01 (For PFS) <0.01 (For OS)
	IM ^3^	142.3		
	Low and IM		166	
	High	110.2	149	
RAI ^4^	Received		193	0.01 (For OS)
	Not Received		155.3	
Distant Metastasis	Absent	173	194.7	<0.01 (For PFS)
	Present	106.5	143	<0.01 (For OS)
CPG ^5^	Applied	161.3		<0.01 (For PFS)
	Not Applied	157		

^1^ PFT: Progression-free time; ^2^ ST: Survival time; ^3^ IM: Intermediate; ^4^ RAI: Radioactive iodine; ^5^ CPG: Clinical practice guidelines.

## Data Availability

The data presented in this study are available on request from the corresponding author. The data are not publicly available owing to privacy.

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
