# Peer review of "Long-Term Survival Analysis and Prognostic Factors of Arabic Patients with Differentiated Thyroid Carcinoma: A 20-Year Observational Study at the King Hussein Cancer Center (KHCC) Involving 528 Patients"

_cancers, 2023, doi:10.3390/cancers15164102_

Round 1

Reviewer 1 Report

Thanks the Editor to give me the opportunity to revise this artI read this article with great interest. 
The manuscript is very well written. The work is well structured, clear in all its parts. The topics covered are consistent with each other. The conclusions are well argued. 
Considering what has been said, I have only a few suggestions:

- The main feature of the research is the evaluation of DTC OS and FS in a specific population (ndr Arabic) which have been less studied compared to other (ie caucasian or Chinese). The aim is obviously remarkable, but for this reason the Authors should be more precise around the ethnicity of studied patients, specifying the different nationality of origin. In fact, the study is based on a single-tertiary-center sample, but we do not know if the treated people are all from Jordan or also from other countries and if there are, in case, any differences among the different nationalities. So please revise accordingly, adding this important information.

- In line 189, Authors state that "32 patients underwent to partial thyroidectomy". Please specify what type of surgery (Lobectomy?) and for what reason the have been submitted to this particular surgery.

- Figure 2 legend. Please add already -not only in the main text-  explanation about what "Panel A" and "B" stand for.

Author Response

Letter to Reviewer 1

  • Thanks the Editor to give me the opportunity to revise this article read this article with great interest. The manuscript is very well written. The work is well structured, clear in all its parts. The topics covered are consistent with each other. The conclusions are well argued.

Thank you for your comments. It was very inspiring to recommend our work on this review and to provide instructive comments that were used to reinforce and strengthen vital aspects of this manuscript.

Considering what has been said, I have only a few suggestions:

  1. The main feature of the research is the evaluation of DTC OS and FS in a specific population (Arabic) which have been less studied compared to other (i.e., Caucasian or Chinese). The aim is obviously remarkable, but for this reason the Authors should be more precise around the ethnicity of studied patients, specifying the different nationality of origin. In fact, the study is based on a single-tertiary-center sample, but we do not know if the treated people are all from Jordan or also from other countries and if there are, in case, any differences among the different nationalities. So please revise accordingly, adding this important information.

Thank you for bringing this important point to our attention. We have updated table 1 to include patients’ nationality in response to your respectful review. Kindly track changes highlighted in grey.

  1. In line 189, Authors state that "32 patients underwent to partial thyroidectomy". Please specify what type of surgery (Lobectomy?) and for what reason the have been submitted to this particular surgery.

This sentence was updated and enriched in response to your respectful review. Kindly track changes (highlighted in grey) in line 188-192.

  1. Figure 2 legend. Please add already -not only in the main text- explanation about what "Panel A" and "B" stand for.

Figure 2 legend has been corrected per your recommendation. Kindly track changes in lines 214-215 (highlighted in grey).

Reviewer 2 Report

I would like to thank Al-Ibraheem et al for their paper in which they tried to gather a large population of Arabic patients with DTC in order to examine their long-term survival and to identify factors that could affect disease progression. 

I would like to point out a few points for the improvement of the manuscript:

Throughout the text there are several abbreviations that have not been analyzed when first mentioned, for example PTC, RAI, DTC, WBS, OS, PFS, WBS, NM and so forth. The same holds true for each figure and table, where each abbreviation should be analyzed irrespective of the text.

Line 37: “disease events”: please rephrase or better specify

Line 64: it should read World Health Organization

Line 73: of the different types of DTC

Line 81: “end up dying from DTC”: please rephrase

Line 90: it should be specified at this point (and not only in the Materials and Methods section) that data were gathered retrospectively

Line 99: “Among this group”: please rephrase

Lines 118 and 134: It would be probably better to avoid abbreviations in the titles

Lines 130 and 132: “were kept” and “was then performed”. Please amend

Line 183: “The mean tumor size was”, please amend

Line 189: “followed by complete thyroidectomy”, please amend

Lines 252, 253: “has no effect” and “gender was linked”, please amend

Lines 254-260: This sentence should be omitted at this point. A more general description of the paper’s results should be included at the first paragraph of discussion, and then each point should be more extensively discussed in the following paragraphs

Lines 282-284: This sentence should be rephrased, probably by omitting Although at its beginning

Line 306: “simulates”, please exchange for another, more relevant, word

The manuscript needs slight improvements mostly discussed in the previous section.

Author Response

Letter to Reviewer 2

  • I would like to thank Al-Ibraheem et al for their paper in which they tried to gather a large population of Arabic patients with DTC in order to examine their long-term survival and to identify factors that could affect disease progression. 

  • Thank you for your insightful comments. It is very motivating to know that our work is appreciated and will hopefully have a positive impact on such an important topic.

  • I would like to point out a few points for the improvement of the manuscript:
  1. Throughout the text there are several abbreviations that have not been analyzed when first mentioned, for example PTC, RAI, DTC, WBS, OS, PFS, WBS, NM and so forth. The same holds true for each figure and table, where each abbreviation should be analyzed irrespective of the text.
    1. Line 37: “disease events”: please rephrase or better specify
    2. Line 64: it should read World Health Organization
    3. Line 73: of the different types of DTC
    4. Line 81: “end up dying from DTC”: please rephrase
    5. Line 90: it should be specified at this point (and not only in the Materials and Methods section) that data were gathered retrospectively
    6. Line 99: “Among this group”: please rephrase
    7. Lines 118 and 134: It would be probably better to avoid abbreviations in the titles
    8. Lines 130 and 132: “were kept” and “was then performed”. Please amend
    9. Line 183: “The mean tumor size was”, please amend
    10. Line 189: “followed by complete thyroidectomy”, please amend
    11. Lines 252, 253: “has no effect” and “gender was linked”, please amend
    12. Lines 254-260: This sentence should be omitted at this point. A more general description of the paper’s results should be included at the first paragraph of discussion, and then each point should be more extensively discussed in the following paragraphs
    13. Lines 282-284: This sentence should be rephrased, probably by omitting Although at its beginning
    14. Line 306: “simulates”, please exchange for another, more relevant, word
  • Thank you for providing in-depth corrections to strengthen the language quality and the point of view shared. We have carefully amended all required changes in line with your respectful recommendations. Kindly track changes in the corresponding lines (highlighted in yellow).
    1. Line 37
    2. Line 64
    3. Line 73
    4. Line 82
    5. Line 90
    6. Line 99
    7. Line 118 & 133
    8. Line 130 & 132
    9. Line 182
    10. Line 190
    11. Line 253 & 254
    12. Removed
    13. Line 255-256
    14. Line 300
